

# A remote-control datalogger for large-scale resistivity surveys and robust processing of its signals using a software Lock-In approach

Frank Oppermann[1] and Thomas Günther[1]

[1]Leibniz Institute for Applied Geophysics, Hannover, 30655, Germany

*Correspondence to*: F.Oppermann (Frank.Oppermann@liag-hannover.de)

**Abstract**

We present a new versatile datalogger that can be used for a wide range of possible applications in geosciences. It is adjustable in signal strength and sampling frequency, battery-saving and can remotely be controlled over Global System for Mobile Communication (GSM) connection so that it saves running costs, particulaly in monitoring experiments. Internet connection allows for checking functionality, controlling schedules and optimizing preamplification. We mainly use it for large-scale Electrical Resistivity Tomography (ERT), where it independently registers voltage time series on three channels while a square wave current is injected. For the analysis of this time series we present a new approach that is based on the Lock-In (LI) method, mainly known from electronic circuits. The method searches the working point (phase) using three different functions based on a mask signal, and determines the amplitude using a direct current (DC) correlation function. We use synthetic data with different types of noise to compare the new method with existing approaches, i.e. selective stacking and a modified Fast Fourier Transformation (FFT) based approach that assumes a 1/f noise characteristics. All methods give comparable results, the LI being better than the well established stacking method. The FFT approach can be even better but only if the noise strictly follows the assumed characteristics. If overshoots are present in the data, which is typical in the field, FFT performs worse even with good data which is why we conclude that the new LI approach is the most robust solution. This is also proved by a field data set from a long 2D ERT profile.

## 1    Introduction

In geosciences there is a high demand of data from observations of different sorts. Particularly, the understanding of processes requires monitoring of various variables with a high temporal resolution. Typical fields are hydrology, geochemistry and geophysics. Consequently, there is a variety of solutions, mostly for very specific purposes. Such data loggers are often installed in regions without power supply and must therefore have low battery consumption. Furthermore, it is beneficial if communication with data loggers to check the functionality or even to adjust settings as preamplification, sampling time or wakeup times.

Autarkic data loggers are very beneficial in large-scale DC resistivity surveys. ERT is a standard exploration and monitoring technology for environmental and engineering problems owing to its high resolution and low cost. The target parameter electrical resistivity and its inverse, electrical conductivity, exhibit high sensitivity to important variables such as groundwater



salinity or clay content. So-called multi-electrode instruments are popular where a multiple wired cable connects a number of

electrodes. Two electrodes are used to inject a current and two other to measure a voltage. By varying these combinations, hundreds or thousands of measurements are easily conducted per hour. However, the use of these multi-electrode systems is limited to rather small layouts, typically in the order of about 100 electrodes with maximum electrode distances of about 10-20 m. This restricts the depth of penetration to a few hundreds of meters.

In order to achieve signal deeper penetration, one can use dipole-dipole experiments, where both current injection and voltage

measurement are realized by dipoles that are small compared to the total layout Alfano (1974). Usually, square wave signals are injected, because the signal can easily be recognized. Alfano (1980) imaged geological structures by dipole-dipole experiments. The longest measured profiles measure up to 22 km length: Storz et al. (2000) imaged fault zones at the German continental deep drilling site KTB and Schütze & Flechsig (2002) conducted such a profile across the flanks of the long valley caldera by using a large-scale dipole-dipole experiment to image fluid flow (Pribnow et al. 2003). Friedel (2000) used

measurements along the flanks of the Merapi volcano to derive 2D subsurface images to locate the magma chamber. Günther et al. (2011) described how a fault zone can be imaged with large-scale ERT and structural information from seismics along a 2.5 km long profile.

However, one can easily extend the technique to three dimensions. Brunner et al. (1999) investigated a Tertiary Maar using a layout of each 24 dipoles in three rings. Schünemann et al. (2007) used a star layout to derive apparent resistivity tensors over

a buried valley. Similarly, Agricola et al. (2017) used such a layout to map the volcanic structure of the Vogelsberg. Flechsig et al. (2010) demonstrated a feasibility dipole-dipole test in a 20x20 km area inside the Eger rift zone, from which a block model could be derived from the sparsely distributed current and voltage dipoles. A surface-downhole ERT survey was used by Bergmann et al. (2012) in the context of carbon dioxide sequestration. Ronczka (2015) used boreholes as long electrodes to investigate inland saltwater intrusion into a 4x4 km wide area endangered by saltwater upconing (Günther et al. 2015),

already using a prototype of the data logger presented here. Before, most large-scale ERT surveys used either high-frequency single-channel seismic data loggers (e.g., Reftek Texan-125) or three-channel magnetotelluric data loggers (Golden et al., 2004). However, the latter provided only a maximum sampling frequency of 8 Hz leading to folding of anthropogenic noise energy into the signal frequency band (typically slightly below 1 Hz) due to aliasing effects. As long cables are lying in the field, an extended amount of manpower must be spent to control whether there are valid voltages recorded or if the cables have

been affected by human or animal activity. Furthermore, the preamplification must be set up before the measurements and cannot easily be changed, thus resulting in a loss of either data or accuracy. We address these two shortcomings by a new data logger that can be controlled by mobile phone communication protocols.

In all those experiments, long time series are recorded and need to be analysed in order to determine the current and voltage strengths. Friedel (2000) compared different approaches and presented a method based on signal stacking using a program

called DCtrap that was applied in some cases (e.g., Schütze & Flechsig, 2002; Flechsig et al., 2010; Bergmann et al. 2012). Other approaches (e.g. Günther et al., 2011; Agricola et al. 2017) use the energy of the signals in the Fourier spectrum to retrieve signal strength. Schünemann et al. (2007) used an inverse method to determine the correlation coefficient between



current and voltage. This approach is already related to the Lock-In approach, which is also used in some multi-electrode resistivity instruments (e.g., 4point light 10W of LGM electronics, www.l-gm.de).

Lock-In detectors working with the principle of phase-sensitive detection, are widely used to retrieve small signals out of a huge noise floor (Meade, 1982; Meade, 1983; Blair and Sydenham, 1975). The signal to be measured is modulated by a reference frequency. The receiver "locks in" to this frequency thus reducing the effect of ambient noise (Scofield, 1994). This results in detection at very low signal-to-noise ratios. In the beginning only analog Lock-In amplifiers were implemented with tubes (Baker 1954, Dereppe, 1961), operational amplifiers (De Marcellis et al., 2007) and application-specific integrated

circuits (ASIC) (Ferri et al., 2001). Nowadays digital Lock-In detectors implemented by discrete circuits (Saam and Conradi, 1998), digital signal processors (DSP) (Sonnaillon and Bonetto, 2005; Proksch, 2006), field programmable gate arrays (FPGA) (Wilson et al., 2015), microcontroller (Bengtsson, 2012) or software (Andersson et al., 2007) are more common. Digital Lock-In detection is more robust compared to analog solutions, particularly the performances at low frequencies is significantly better. The field of applications, which require the detection of very low signals in a noisy surrounding, where Lock-In

detectors are used is wide spread from optics (Andersson et al., 2007; Masciotti et al., 2008; Holzman et al., 2005), impedance spectroscopy (Albertini and Kleemann, 1997), wireless networks (Gabal et al., 2010), biologic applications (Ferri et al., 2001; Johnson et al., 2001), Electron Spin Resonance (ESR) (Vistnes et al., 1984; Murányi et al., 2004) to Nuclear Magnetic Resonance (NMR) (Saam and Conradi, 1998; Caracappa and Thorn, 2003).

In the following, we are describing layout and usage of the data logger before presenting a new approach for processing the

retrieved time series along with established routines. We use synthetic noisified data to compare the performance of the methods and show the application to field data of a large-scale ERT survey before drawing some conclusions.

## 2  Datalogger

### 2.1.  Hardware

The core of the datalogger (Figure 1) is a GigaLog-S (GigaLord, France) data acquisition module. It contains a 16-channel 24-bit Delta-Sigma A/D converter connected to an Atmel microprocessor. Only 3 of the channels are used by 3 DC-coupled high-impedance differential preamplifier with 4 programmable gain stages (factors 2, 10, 20 and 100) that are controlled by the microprocessor. The sample interval can be chosen between 1 millisecond and 1 second in steps of 1 millisecond. The sampled data will be stored in a text file on a micro SD card. The memory consumption is about 100MB/h at 1ms sample interval. The

datalogger can be accessed through local USB cable or the combined GSM/GPS module Aarlogic C05/4 remote via General Packet Radio Service (GPRS) connection. The core of this module is a Telit GE864-GPS chip. The Global Positioning System (GPS) receiver of this module allows synchronizing the microprocessor clock, setting the time of the realtime-clock and saving the GPS position of the datalogger on the micro SD-card. The system is powered by an external 12V accumulator. The power consumption is about 300mA with active GPRS connection, 150mA with idle GPRS connection and 15mA in sleep mode.

The programmable sleep mode turns the logger system into a power saving standby after operation hours. At a specified time, it is powering up again, restoring all parameters and doing a new GPS time synchronization. Figure 2 shows the system, with



the datalogger connected to the combined GSM/GPS antenna, a 12V accumulator and a laptop with GSM modem and antenna to remotely control the datalogger from a distant place.

## 2.2.  Graphical User Interface (GUI)

The Software GUI is written in LabView. In the single-channel mode you can connect to one logger and control the different functions, like starting or stopping the data acquisition, setting filename, gain or sampling interval. It is possible to read the directory of the SD card, monitor the accumulator voltage, the temperature of the logger or the free memory of the SD card. You can download whole data files, but this is only advisable with USB connection due to the slow transfer rate of the GPRS connection. Another mayor feature of the remote controllability is that you can monitor the signals of the acquired data in realtime. To control many loggers simultaneously we provide a tabular control (Figure 3).  For a survey setting you can choose in the first 4 columns the loggers by filename, gain and sample interval. By executing this sequence every logger will be checked for its online status and the parameters will be set and read back. It is checked whether data acquisition is running, if logger temperature, accumulator voltage and SD memory are within predefined values. A small amount of data is downloaded and checked if the input signal of the 3 channels are in the proper range of amplitude and offset voltage. It will be checked if the datalogger time has been recently synchronized with GPS time. GPS coordinates are displayed and a Google map image is created to show the datalogger positions. Important is also to set the shutdown and wakeup times for the sleep mode.

Figure 4 shows a sample signal containing the response of an injected 0.2 Hz ERT square wave current as visible in the GUI monitoring software. A Fast Fourier Transform (FFT) algorithm calculates the corresponding frequency spectrum. On top of the underlying broadband noise one can see the 50 Hz signal of the powerlines, the 16.7 Hz from the railway lines, and the 0.2 Hz ERT signal with their odd harmonics.

## 3   Signal processing of ERT signals

The most common approach to retrieve the amplitude of an ERT square wave signal from the recorded time series is a stacking algorithm, as used by some multi-electrode instruments (e.g., RESECS by http://www.geoserve.de/). Other instruments (e.g., "4point light 10W" by http://www.l-gm.de) use (hardware) Lock-In amplifiers, which show very good results at very low signal-to-noise ratios. Third, signals can be analyzed in the Fourier domain by relating the FFT energy of both current and voltage spectrum (e.g., Agricola et al., 2017). We process the recorded time series using all three approaches, including a newly developed Lock-In method. To validate the quality of the latter approach, we compare its results with stacking and FFT algorithms. As there are changes in implementation, we describe and illustrate all three variants in the following.  We assume a square wave signal of 0.2 Hz for synthetic studies. This is also used in the field as a trade-off between stability and noise impact: A half period of 2.5 s provides the enough samples for a robust amplitude determination. Higher frequencies would reduce the amount of samples so that the unstable part of transient effects can dominate. Smaller frequencies would probably lead to smaller signal-to-noise ratios due to the typical 1/f noise characteristics.




### 3.1. FFT

For the processing we use the frequency spectrum of the recorded time series. This power spectrum is created by a Fast Fourier Transformation FFT using a Gaussian window. If the duty cycle of the square wave signal is 50%, the frequency power spectrum is:

$$P(t) = \frac{4}{\pi} \sum_{n=1}^{\infty} \left( \frac{\sin(2\pi(2n-1)ft)}{2n-1} \right) \tag{1}$$

We focus on the 1$^{st}$ harmonic of the ERT signal. However, we cannot just take the value at the 1$^{st}$ harmonic as broadband noise contains also significant values at this frequency. To determine the effect of this noise, we have to subtract the noise level around the 1$^{st}$ harmonic (Figure 5). To calculate the broadband noise level we cut out all harmonics of the ERT signal and consider only the frequency range from DC to the 15$^{th}$ harmonics. Unlike Günther et al. (2011) or Agricola et al. (2017) we do not assume a constant noise floor over the frequency axis, but assume a 1/f decrease that is often observed (Surkov and Hayakawa, 2007). Therefore we carry out a nonlinear fit with a noise model consisting of a noise floor and an inverse f contribution:

$$u(f) = a0 + \frac{a1}{a2+f} \tag{2}$$

The peak amplitude Up of the ERT square wave signal is then calculated by the following Eq. (3) from the peak value Po and the noise value Pnoise at the 1$^{st}$ harmonic.

$$Up = \sqrt{\frac{4}{\pi}(Po - Pnoise)} \tag{3}$$

The quality criterion is the signal-to-noise ratio at the base frequency.

### 3.2. Stacking

There are many different ways to stack a signal (Naess & Bruland, 1985). To keep this comparison simple, we used only one stacking method, the alpha-trimmed-mean stack (Friedel, 2000). First step is a drift correction (Friedel, 2000). The drift-corrected function Udr (t) is the time series U (t) subtracted by the moving mean value of the time series with a window size of the period M of the ERT signal.

$$Udr(t) = U(t) - \frac{1}{M} \sum_{d=-M/2}^{M/2} U(t+d) \tag{4}$$

This gives only a correct result if the signal is symmetric which it is mostly the case. At every stack point represented by one sample within the signal period, the median values of all stacks are sorted by amplitude. To eliminate outliers, a variable percentage of the largest and smallest amplitudes is rejected (Figure 6). A rejection (alpha-trim) rate of 10% shifts the distribution into a linear behavior so that the mean is less sensitive to outliers. As we start the stacking at a random point we do not know the correct phase relation yet. A maximum cross correlation between the stacked signal and an ideal signature will usually point to the leading and tailing edge. We increase the sensitivity by doing the cross correlation for all possible phase relations and pick the phase at the highest cross-correlation coefficient. To remove effects like overshoots at the slope of the ERT signal, the average value will be calculated in a variable window of the signal plateau (Figure 7). The main quality criterion is the ratio of the average values $U_{pos}$ at the positive and $U_{neg}$ at the negative plateau. Secondary quality criteria is the



maximum cross correlation coefficient and the mean-square-error (MSE) of the signal at the plateau compared to the average value.

170

### 3.3. Lock-In

In the Lock-In processing, also known as phase sensitive rectifying, the negative part of the acquired square wave is rectified, accomplished by a reference signal at the slopes of the square wave. Harmonic noise annihilates itself and disharmonic noise is thus reduced. The DC value of this convoluted signal is the desired amplitude. Usually Lock-In amplifiers retrieve the reference signal hard-wired from the transmitter in order to have the correct phase information at the receiver. In a few square kilometer large survey area it is not practical to connect all datalogger with trigger cables to the current source, thus destroying the advantage of the autonomous remote-controlled datalogger. However, we show that it is also possible to solve this problem numerically.

First step is a drift correction (see 3.2. Stacking). With a sample interval of N milliseconds and a period of M milliseconds of the injected current we have a number of $P = M \times N^{-1}$ possible phase relations between the acquired and the reference signal. If we generate an artificial numeric reference signal M (t) and convolute this with the time series (Figure 8) of the received drift-corrected signal Udr (t) for every phase relation i=0 to P we get the convoluted signal Uc Eq. (5).

$$Uc(t, i) = Udr(t) * M(t + i) \text{ if } M(t + i) <> 0 \tag{5}$$

185

To remove effects of overshoots at the slopes, the reference signal M(t) has, additionally to the usual states "+1" and "-1", a third state "0". When M (t) =0 the values of the received signal Udr(t) are not included in the convolution. By calculating the DC value DC(i) Eq. (6), the peak-to-peak voltage Vpp(i) Eq. (7) and effective value RMS(i) Eq. (8) of the convolution Uc(i) we obtain 3 functions (Figure 9) from where it is possible to derive the correct phase relation.

$$DC(i) = \frac{1}{N}\sum_{t=0}^{N} Uc(t, i) \tag{6}$$

$$Vpp(i) = \max_{0 \le t \le N} Uc(t, i) - \min_{0 \le t \le N} Uc(t, i) \tag{7}$$

$$RMS(i) = \sqrt{\frac{1}{N}\sum_{t=0}^{N} Uc(t, i)^2} \tag{8}$$

The main criteria to get the real phase is the maximum of the DC function, if within a specific DC search area (THdc threshold of DC function, usually 25%). As at certain additional signals like overshoots this can lead to wrong results, we have to look if there is also a minimum of the Vpp function within a Vpp search area (THvpp threshold of Vpp function, usually 20%). A minimum of the RMS function below the RMS search area (THrms threshold of RMS function, usually 10%) will improve



accuracy. This procedure is shown in the flow diagram in Figure 10 (a) and shown on the example in Figure 10 (b): The minimum of the quadratic sum *eval(i)* of the 3 normalized criteria functions determines the phase, whereas we find the amplitude in the DC function DC(i).

The convolution of the acquired square wave signal with the reference square wave signal (mask signal in Figure 8) results in a triangle shape function DC(i). The linearity of the triangle slope is correlated to the quality of the acquired square wave signal shape. The mean square error MSE between the triangle signal to a fitted straight line is a good quality criterion for the Lock-In result. We used only the area between 20 and 80% for the fit to avoid errors caused by overshoots. The MSE can also be used to find the perfect value of the "0"-state. To find the best cutting point of transient effects and to determine the plateau of the signal it is advisable to do the convolution with different percentages of the M (t) = 0 state and see where the MSE has a minimum (Figure 11.). At a "0" state of about 25% the MSE has its minimum. The signal is stable at 10 mVp at the plateau from "0"-state 16 to 25%. Below 16% it does not cut the overshoot appropriately. Too high percentages of the "0"-state lower the sensitivity of the Lock-In detection because the noise reduction effect of the method decreases.

## 4    Quality assessment of the processing methods

### 4.1        Synthetic data

To compare the results of the Lock-In-, stacking- and FFT-processing we created two artificial datasets. Both datasets exhibit a square wave signal with a frequency of 0.2 Hz using 50% duty cycle and an amplitude of 10 mVp. The first dataset has a 10 mV overshoot (5% duty cycle wide) at the leading and trailing edge of the signal added. The second dataset is without overshoot. To create realistic noise conditions, we added 75 mV railway noise at 16.7 Hz and 100 mV power noise of 50 Hz. The variable noise source was a pink noise with increasing effective noise amplitude from 0 to 249 mVrms added to both signals. The pink noise is created from white noise filtered through a 1/f digital filter with 0.1 Hz and 100 Hz lower and upper cutoff frequency, respectively. Each value is the average of 200 processing results with different seeds for the random generator.

### 4.2   Comparison of methods

Each processing has different kind of quality criteria to judge if a result is valid or not. At the Lock-In it is the mean-square-error of the slope of the DC-function. At the stacking it is the ratio of the positive and the negative plateau amplitudes and at the FFT it is the spectral signal-to-noise ratio at the base frequency. For the comparison we adjusted the threshold values of the different quality criteria so that an equal number (30%) of invalid results were rejected. This leads of course to an overestimation towards higher noise values of all three methods, because at low amplitudes the tendency is higher that they will be rejected by the quality criteria. Figure 12 shows the 3 different quality criteria over noise amplitude in a normalized graph. The criteria for FFT, Lock-In and stacking show an exponential increase. Figure 13 shows the results of the comparison of the three processing algorithms for the two datasets without (a) and with (b) overshoot. For low noise amplitudes all methods



yield similar results except the FFT at the dataset with overshoots, because at the frequency spectrum we cannot separate between the fractions from the ERT signal and the overshoot. Lock-In and stacking cut off overshoots at the signal slopes and are therefore not sensitive to transient effects if the "0"-state has been chosen appropriately (typically 20%). At higher noise amplitudes stacking is overestimating the signal more than Lock-In and FFT. Stacking bases the amplitude decision on a wider frequency spectrum, while Lock-In and FFT are just looking at a narrow frequency band. Lock-In and FFT are therefore less affected by broadband noise. Note that the performance of FFT might be overestimated as we applied a noise model that perfectly matches the 1/f behavior in the FFT approach.

### 4.3   Relation of quality factor to S/N

The quality criteria MSE of Lock-In data of the synthetic data over noise can be used to validate the quality of the processed data. If we assume the function of MSE over noise (Figure 12) valid for realistic signal amplitudes we can normalize the signal-to-noise ratio with the signal amplitude of 10mV:

$$MSE = \text{f(noise)} = \text{f}(S/N) \tag{9}$$

Then we fit this function Eq. (9) with an exponential fit, because this function describes the relation sufficiently well in the relevant S/N range:

$$\text{MSE} = a * e^{-(b * S/N[dB])} \tag{10}$$

Now it is possible to calculate the signal-to-noise ratios of data from the matching MSE values with the fitting results a=0.0271 and b=0.2949. This means that for a zero signal ($S/N$ of 1 equaling 0 dB) we obtain an MSE of 0.027 and an increase to about 10 for $S/N$ of about -20 dB (S/N of 0.1). For any MSE we can thus reconstruct $S/N$ by rearranging Eq. (10):

$$S/N [dB] = -\frac{1}{b} * \ln\left(\frac{MSE}{a}\right) \tag{11}$$

### 5   Field case application

We apply the developed data logger and analysis methods to a large-scale ERT field case measured near the town of Schleiz in Thuringia, Germany. We focus on the second part of the field campaign and of the part of the profile and remove data from two other data loggers. The slightly reduced profile thus consists of 31 electrodes with a spacing of 125 m. We used a high-current generator to maximize the magnitude of the measured voltages. Into each dipole we injected the maximum current ranging from 3A to 25 A depending on coupling conditions. As for close dipoles the measured voltage exceeded the input range, we additionally injected a small current of about 2-3 A. Here we only used these data for reasons of homogeneity and the greater S/N range.

A complete dipole-dipole survey was carried out. The principle of reciprocity states that interchanging current and dipole does not change the measured impedance (Friedel, 2000). As a consequence, reciprocal measurements can be used for error





estimation. For single data this is a check for manual inspection to decide how the two independent measurements (a forward and a backward dipole-dipole) are combined before going into the inversion routine. For multi-electrode data it has become a common practice to use reciprocal data in a statistical sense to derive error models for weighting data in inversion (Udphuay et al., 2011). Such an error model consists of a constant relative error and a voltage error to take the magnitude of the data into account. It is achieved by distributing the data into bins of similar voltage and fitting a curve to the standard deviation of the

reciprocal error.

For estimating the error level we use Eq. (11) to calculate the S/N of every data point using the matching quality factor MSE. If we assume for example an amplitude error of >5% is not acceptable, we see in Figure 13a that the errors start growing above 5% at a noise level of 100mVrms. This results at a signal amplitude of 10mV to a S/N of -20dB. Figure 14a shows the distribution of the signal-to-noise ratio in comparison with the MSE of the LI processing of the synthetic data. The majority

of the field data are in the region where the MSE and therefore the amplitude errors are negligible. Just some data below the critical S/N of -20dB have an MSE >10 corresponding to amplitude errors >5%. Figure 14b shows the obtained S/N for forward-directed standard dipole-dipole data in a so-called pseudo-section, i.e. as a function of the array midpoint and the dipole separation factor DD (dipole distance by dipole length). Higher DD correspond to larger penetration depths but exhibit lower measured voltages and thus generally worse S/N.

All three processing approaches were applied to the time series. The resulting voltages were divided by the driving current and multiplied by the analytically known geometric factor resulting in the so-called apparent resistivity representing the resistivity of an equivalent homogeneous half-space. Additionally, we computed from each available pair of forward and backward measured array the normal reciprocity, i.e. the percentage difference divided by the mean value. Figure 15 shows the pseudosections of the apparent resistivity (mean values of forward and backward measured) and reciprocity for the three

approaches. The upper images are hardly distinguishable, only a few points show distinct deviations due to the wide range of values between 1 and 1000 Ωm. Only some single points are significantly different from their neighbors. There are only a few values missing for the FFT approach in the area of lowest values (and S/N). This proves that all methods are generally applicable. The lower row shows the detected reciprocity. There is some systematics that might result from a non-match of current electrodes (large coupling fields) and potential electrodes (small Ag/AgCl pots). Very few values are below (light grey)

or above (dark grey) the given color limits. The FFT methods has the largest gaps (least number of reciprocal pairs) and overall the largest absolute values, whereas the Lock-In image shows the smoothest image (similar to stack). There are strong similarities between the reciprocity and the derived S/N pseudosection (Fig. 14b) indicating that the MSE error is able to deliver uncertainty information.

**6    Conclusions**

Remote dataloggers can provide valuable information on the subsurface in multi-source or monitoring experiments. The presented datalogger, just powered by ordinary batteries, can register long time series with sampling rates of up to 1 kHz. It



can be accessed and controlled remotely using GSM connection and can therefore save time for a lot of different geoscientific experiments, e.g. in the fields of environmental and groundwater monitoring, or in applied geophysics.

One example are large-scale ERT surveys for geological investigations that cannot be done with conventional instruments. The registered voltage time series have to be processed to obtain single voltages and their measuring uncertainty. For the ERT experiments that typically use square waves with periods below 1 s and 50% duty cycle, there have been two approaches for data analysis, i.e. stacking and FFT methods. In addition, we present a Lock-In based approach, a technique that had been used as hardware solution but can also be applied numerically. To make it robust against different kinds of transient effects and

noise, several functions are computed from the convolution with a mask signal. The combination of them is used to determine the working point (phase lag), the voltage amplitude and a measure of uncertainty based on a mean square error. The software Lock-In processing represents a robust method for determine signal the strength using a reference signal and is thus well suited for ERT.

   Synthetic data are used to compare the new method (LI) with an alpha-trimmed stacking and an improved FFT based method

that includes a 1/f noise model. All methods are able to retrieve correct voltages down to S/N ratios of about 1/10. In all cases, LI performs slightly better than stacking. The new FFT method can sometimes perform even better than both LI and stacking, but this is due to the fact that the synthetic noise was perfectly *1/f* which is not true in reality. Moreover, the often occurring overshoots affect the FFT method even at very good S/N, which is why we consider LI as the most robust method for voltage determination. Moreover, LI provides a good quality criterion (MSE) that can be transferred into S/N or a relative error for

inversion. The applicability could be proved on a large-scale ERT data set and checked using reciprocity analyses. In general, all three approaches provided results that are almost identical. Additionally, The Lock-In approach provides an error measure giving an idea about estimating accuracy of the individual data and resembles the estimates from reciprocity. It needs to be checked whether this measure can be used for data inversion, either individually or in a statistic sense.

   There is space for future development of both the dataloggers and the processing schemes. One could analyze the decay of the

voltage curves to retrieve induced polarization properties. However, this range is often dominated by overshoots, hence IP analysis would require exact knowledge of the source signals. Beyond direct-current (frequencies of 1 Hz or below) one could use the instrument for controlled-source electromagnetic (CSEM) surveys that use frequencies from several Hz to several kHz. However, the limited sampling rate of 1 kHz restricts the frequency range and would require a new logging concept. The Lock-In approach (and similarly stacking) is expected to work until a sufficient amount of samples are required. Therefore we expect

a better performance of FFT for higher frequency, however this is very sensitive to the 1/f noise characteristics and requires discrimination from the powerlines higher harmonics.



*Author contribution*

F.Oppermann developed the datalogger including the control software, worked on the processing methods and prepared the manuscript with contributions from the co-author. T.Günther analyzed the field data and put the manuscript into the context of ERT.

*Competing interests*

The authors declare that they have no conflict of interest.

*Acknowledgements*

We like to thank Raphael Rochlitz, Robert Meyer, Dieter Epping and Vitali Kipke to help producing the field data as part of the DESMEX project funded by the German Ministry of Education and Research (BMBF) under grant number 033R139D.

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






**Figures**

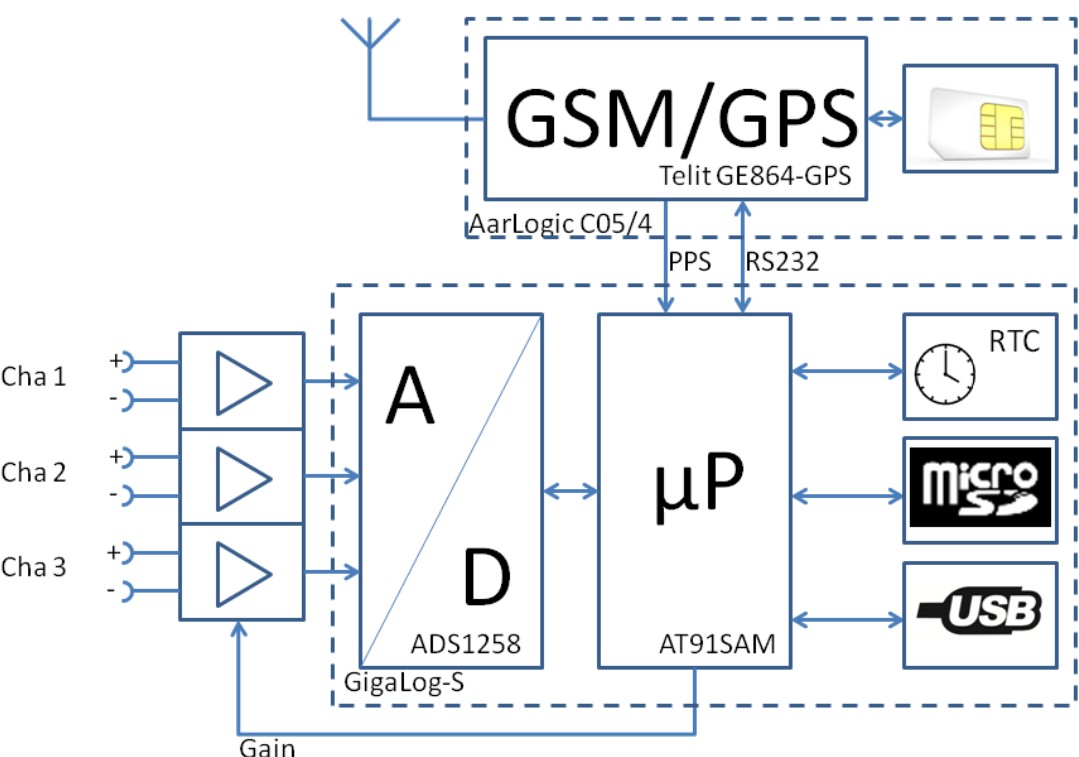

**Figure 1: Block diagram of datalogger consisting of a GigaLog-S data acquisition module, a combined GSM/GPS modem and 3 preamplifiers.**



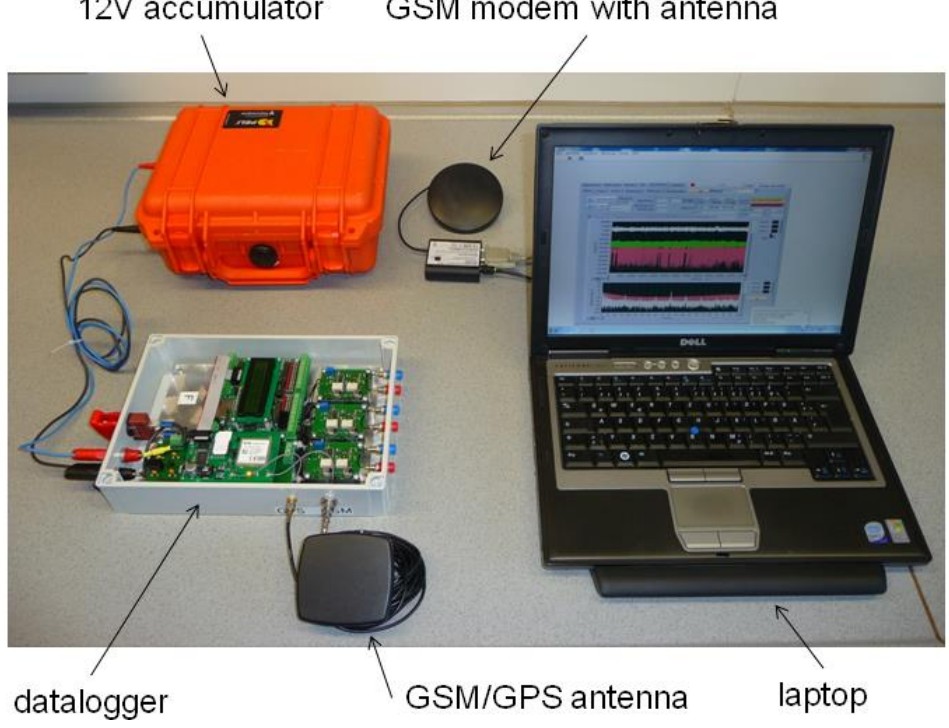


**Figure 2: Datalogger system with datalogger, GSM/GPS antenna and 12V accumulator. A laptop with GSM modem and antenna to remote control.**





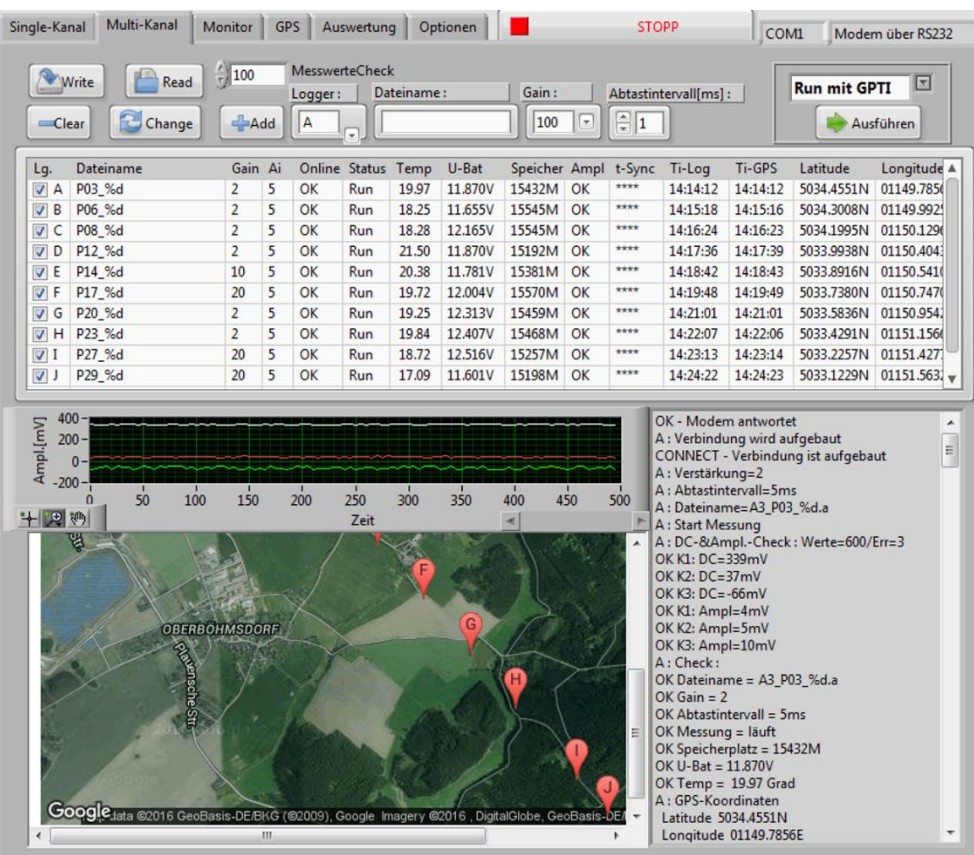

**Figure 3: Software GUI in multi-channel mode**



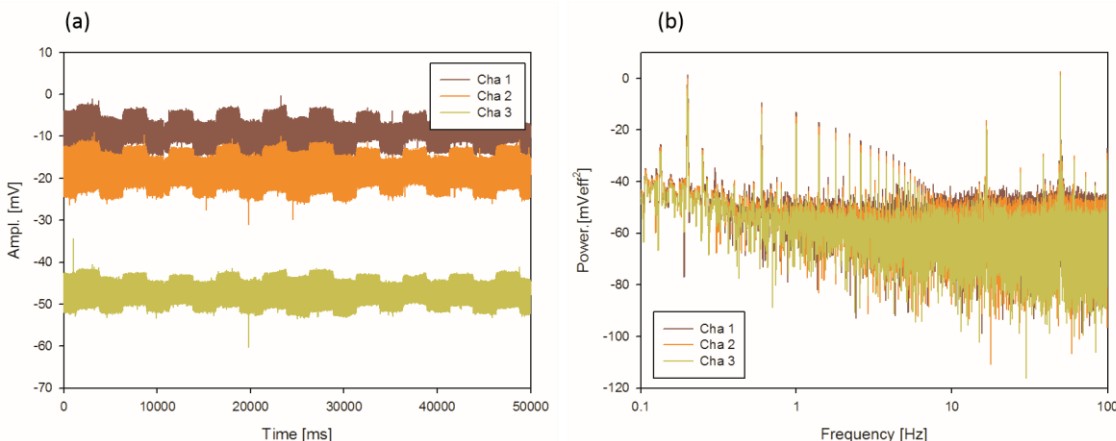

**Figure 4: Exemplary voltage time series (a) from a square-wave current injection and frequency spectrum (b) as visible in the GUI monitoring software.**




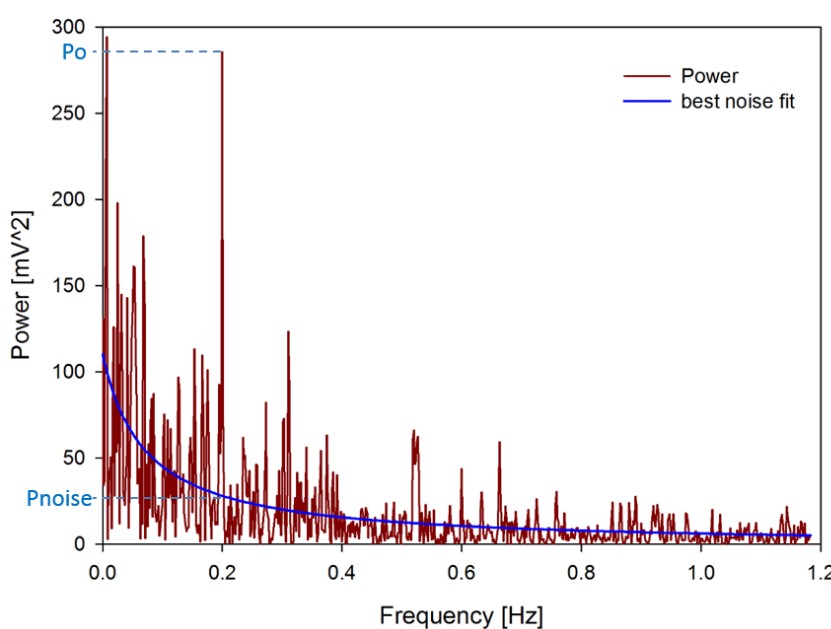

**Figure 5: FFT-based fitting of a noise function (Pnoise) and calculation of the signal strength (P=Po-Pnoise) at the 1st harmonic f=0.2Hz.**




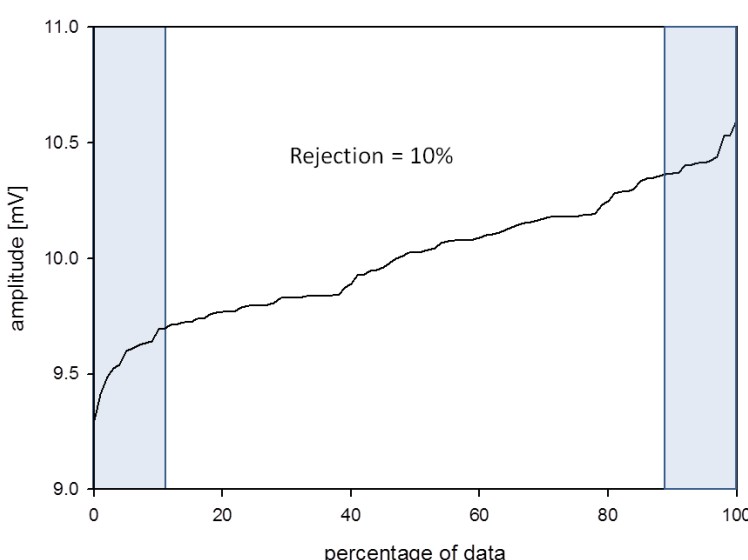

**Figure 6: Sorted amplitude distribution for the alpha-trimmed-mean stacking.**



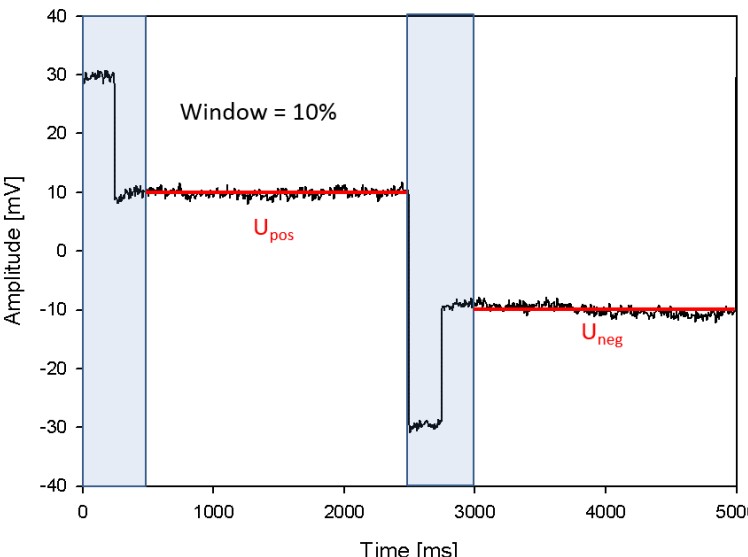


**Figure 7: Determination of effective stacking window for calculating the mean value at the plateaus for a synthetic signal containing a 250 ms overshoot.**


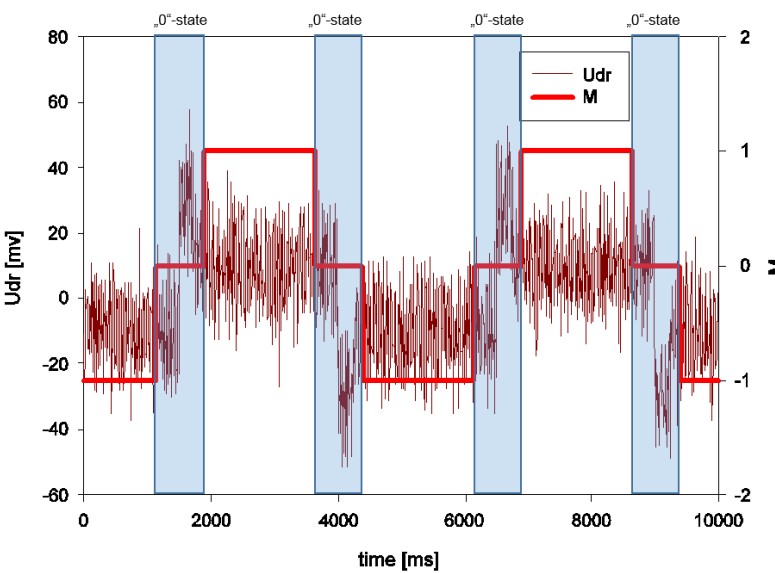


**Figure 8: Drift-corrected signal Udr(t) and mask signal M(t) in correct phase.**



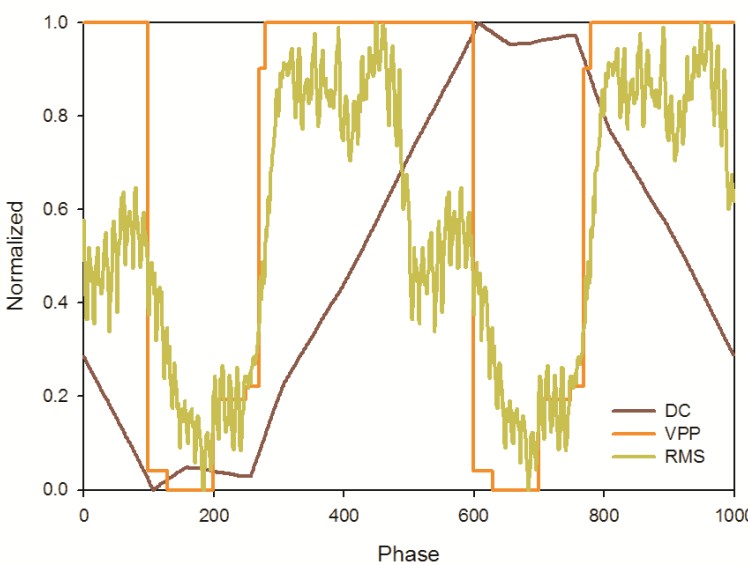

**Figure 9: Evaluated functions Direct current (DC) value, peak-to-peak voltage Vpp and effective root-mean square (RMS) of the mask-convoluted signal.**





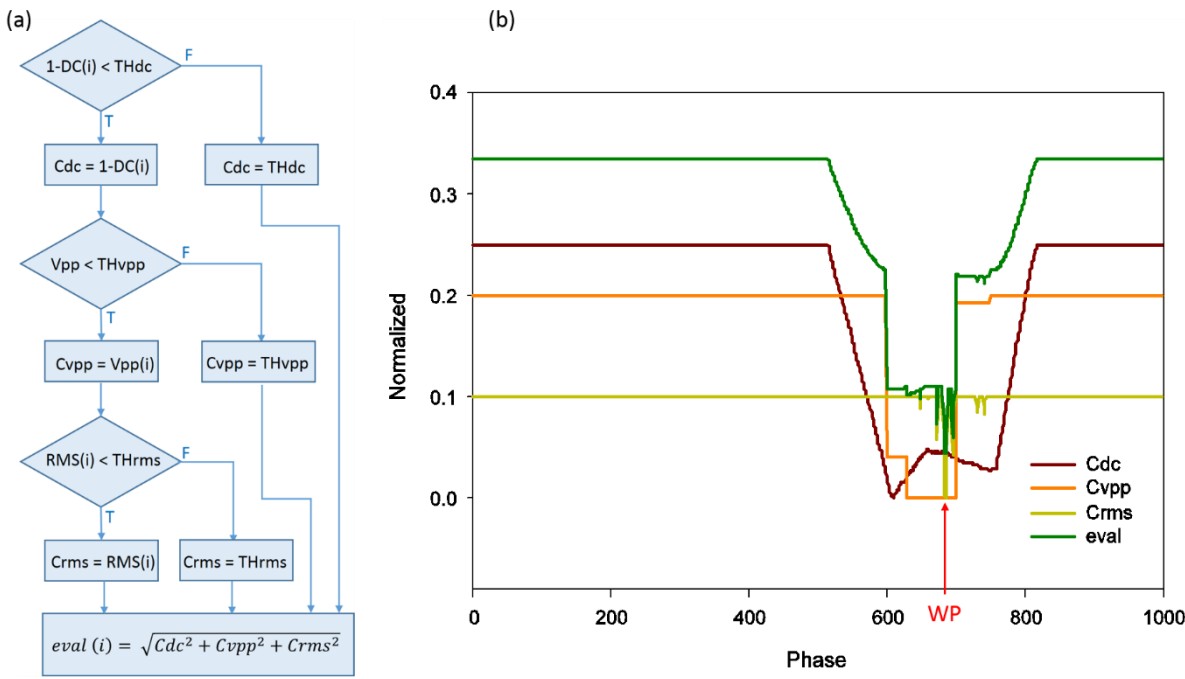

**Figure 10: (a) Flow diagram calculation the components Cdc, Cvpp and Crms from the three functions. (b) Components and quadratic sum for evaluating the working point WP from its minimum.**






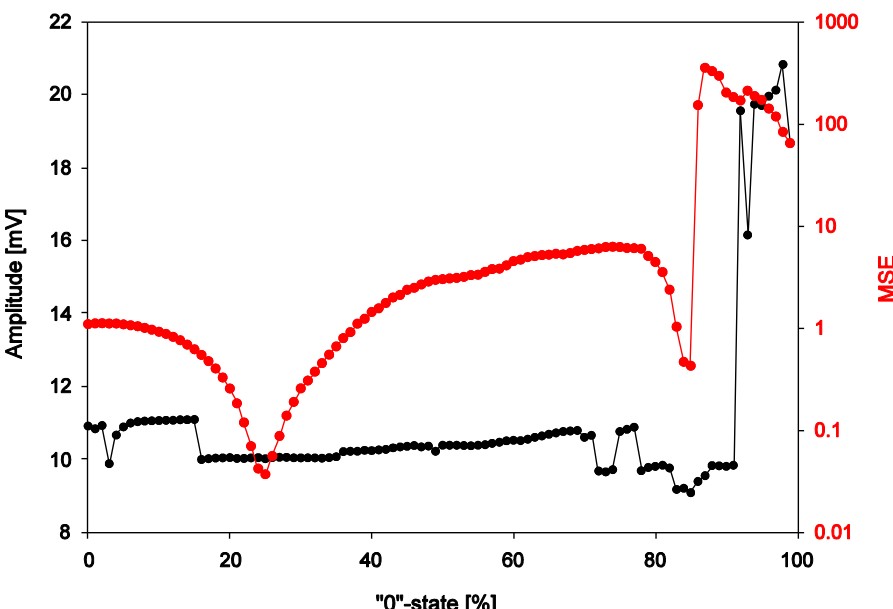

**Figure 11: Detected amplitude and mean square error MSE of a artificial signal with 10mV overshoot and 29mVrms pink noise over length of "0"-state, showing the plateau between 16 and 35%. Up to 16% "0"-state is the effect of the overshoot, above 70% the reduced noise rejection.**



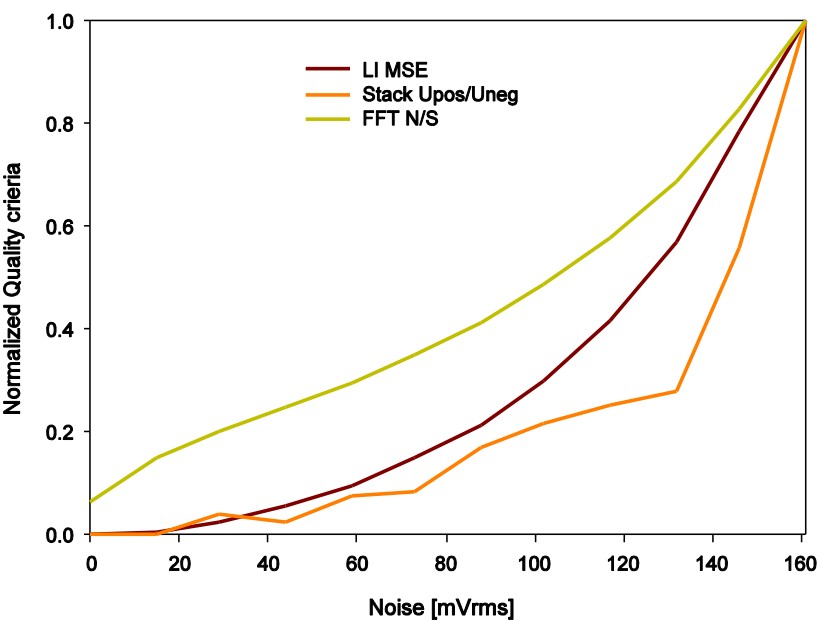


**Figure 12: Quality criteria over noise level of test dataset without overshoot.**



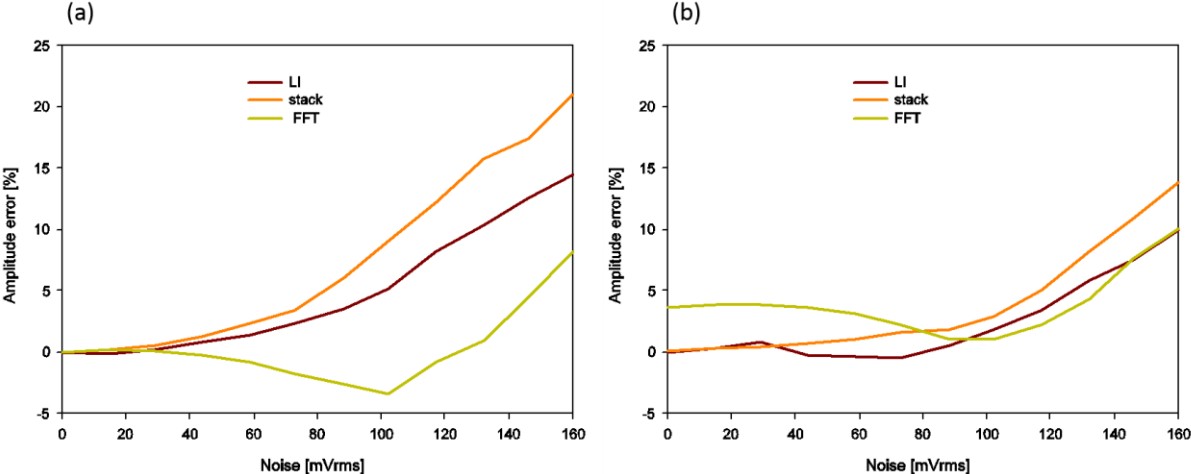

**Figure 13: (a) Comparison LI, Stack and FFT over pink noise without overshoot, (b) Comparison LI, Stack and FFT over pink noise with overshoot.**



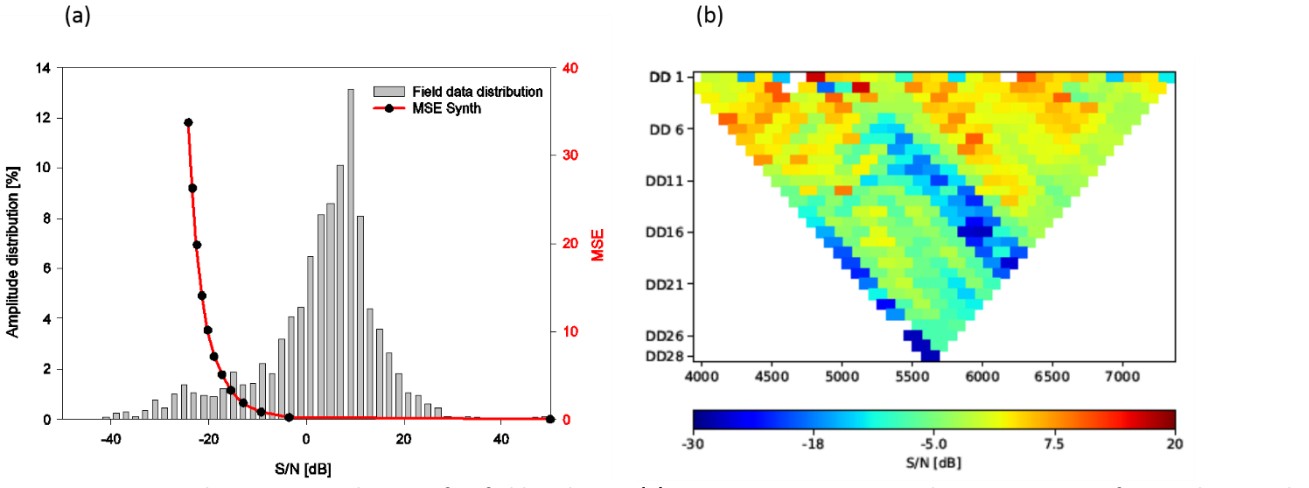

**Figure 14: Signal-to-noise values of field data (a) Histogram compared to MSE of synthetic data, (b) in a pseudosection**





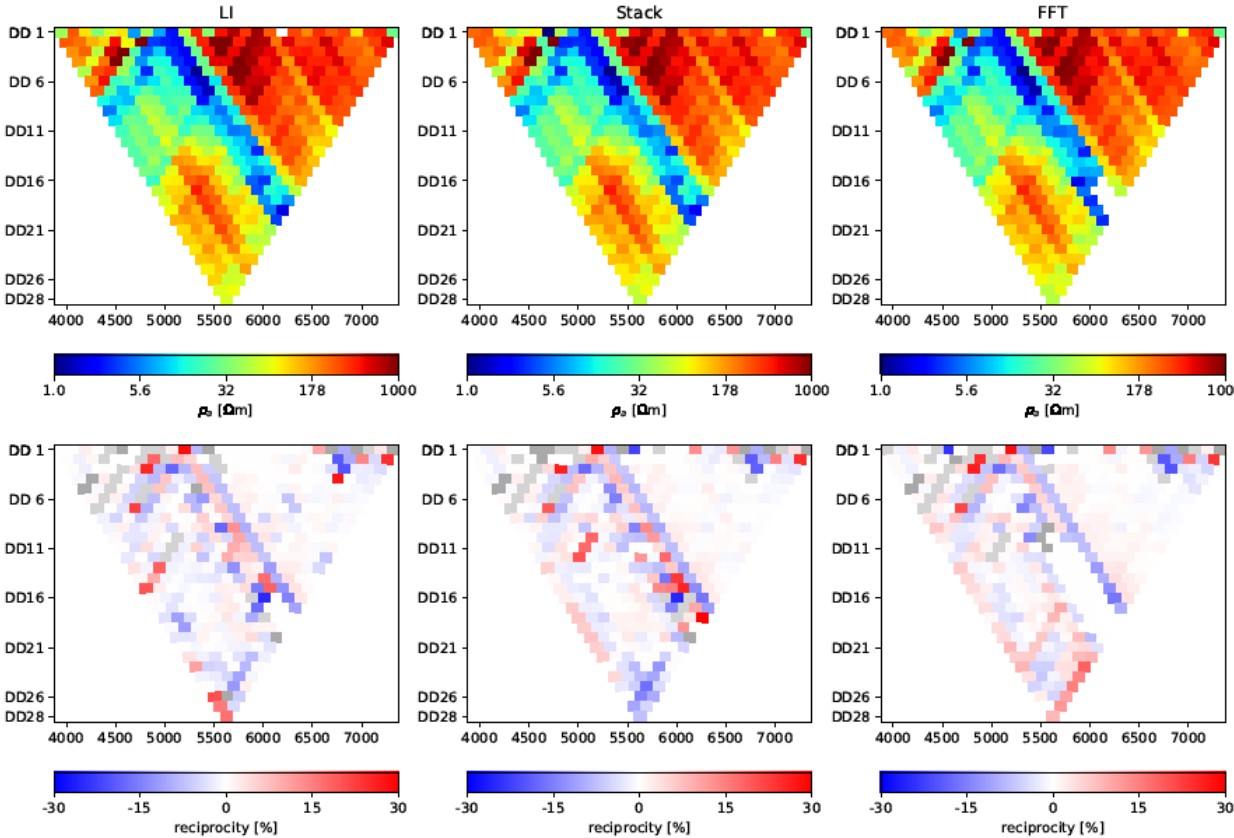

Figure 15: Calculated apparent resistivity (top) and normal reciprocity (bottom) for the three analysis approaches Lock-In (left), Stacking (center) and FFT (right)