# Peer review of "A remote-control datalogger for large-scale resistivity surveys and robust processing of its signals using a software Lock-In approach"

_Geoscientific Instrumentation, Methods and Data Systems, 2017_

## Referee Comment (RC1) · Anonymous Referee #1 · 13 Oct 2017

Comments to editor

The paper is very interesting for people works on geoelectrical no-standard system for deep investigation. The introduction misses some papers on the DC deep approach that already use no-standard equipment. Moreover, the authors introduced the Lock-in approach a methods on the correlation between current and voltage signals and it is used when there are small signals out of a huge noise floor. Anyway, the paper introduce a digital Lock-in detection, that is considered more robust then the analoge one. Finally, I think that this approach adds new things and I suggest to publish it.

Comments to authors

Introduction: The introduction is well described but some cited papers are not indicated in the final bibliography. Moreover, I suggest to add some deep DC application with no-standard instruments (transmitter and receiver physically separated) with a single and multichannel system. In example, there are papers where a deep DC instrument with single channel was used: a)Rizzo E., Colella, A., Lapenna, V. and Piscitelli, S. (2004). "High-resolution images of the fault controlled High Agri Valley basin (Southern Italy) with deep and shallow Electrical Resistivity Tomographies". Physics and Chemistry of the Earth, 29, 321-327;

b)Colella A., Lapenna V., Rizzo E. (2004). High-resolution imaging of the High Agri Valley basin (Southern Italy) with Electrical Resistivity Tomography. Tectonophysics, 386, 29-40;

c)Tamburriello G., M. Balasco, E. Rizzo, P. Harabaglia, V. Lapenna, A. Siniscalchi. Deep electrical resistivity tomography and geothermal analysis of Bradano foredeep deposits in Venosa area (Southern Italy): first results. Annals of Geophysics, Volume 51,No.1, pag.203-212, February 2008)

Moreover, there are some more recent with deep DC multichannel use: a)Santilano A, Godio A, Manzella A, Menghini A, Rizzo E, Romano G (2015). Electromagnetic and DC methods for geothermal exploration in Italy, state-of-the-art, case studies and future developments. First Break . First Break 33 (8), 81-86

b)Balasco M., Giocoli A., Lapenna V., Rizzo E., Romano G., Siniscalchi A., Votta M. (2008). Deep resistivity image of the Agri Valley (Southern Italy). Near Surface 2008 – 14th European Meeting of Environmental and Engineering Geophysics Kraków, Poland, 15 - 17 September 2008

Datalogger: Line 90 to 92: I suggest to explain why only 3 channels are used.

Line 96: The GSM module is used only for the communication between the DL and the

laptop as remote control. . .to download the acquired data the system uses a USB way. Why is it not possible to use a 3G module? I suggest to add one sentence to explain.

Line 120: I suggest to add on the figure 4 the frequencies indicated: powerlines, railway, the signals 0.2hz and the harmonics.

Field case: Line 275-279: the authors wrote "Higher DD correspond to larger penetration depths but exhibit lower. . .". . .the figure 14b show low S/N signals (blue color) in two zone (one shallow and one deep) with in the middle a better S/N signals zone. Therefore, the sentence needs some more details. . . it depends also for the electrical resistivity distribution. Low resistivity zone (i.e. clay) produce low S/N signals then relative high resistive layer (i.e. sandstone). I suggest to explain better this part.

References: I suggest to check the matching between the list of the references and the indication in the text.

Figures: Figure 15: I suggest to add the unit (may be "m") for the X axis

––––––––––––––––––––––––––––––

---

## Author Comment (AC1) · 10 Nov 2017

Dear referee,

thank you for reviewing our manuscript. We have considered your helpful remarks and included the necessary additional information in the manuscript:

1.       Introduction: The introduction is well described but some cited papers are not indicated in the final bibliography. Moreover, I suggest to add some deep DC application with nostandard instruments (transmitter and receiver physically separated) with a single and multichannel system. In example, there are papers where a deep DC instrument with single channel was used: Rizzo E... (2004), Colella A… (2004), Tamburriello G… (2008). Moreover, there are some more recent with deep DC multichannel use: Santilano A… (2015), Balasco M… (2008)

Response: Thank you very much for this suggestions to literature that we have not been aware of. We extended the introduction and parts of the processing with references to these works.

New text: "Furthermore, an Italian group used deep electrical resistivity tomography (DERT) to successively image several deep geological structures (e.g. Santilano et al., 2015 for an overview). Colella et al. (2004) images the Agri sediment basin on profiles of up to 6.5 km length. Rizzo et al. (2004) compare these results with small-scaled ERT. Tamburiello et al. (2008) revealed geothermally relevant fluid-affected structures. Balacso et al. (2011) give insight into tectonic structures in the area of the disastrous L'Aquila earthquake."

Additionally we cited Balasco et al. (2011) as example for a data logger with a similar (but simpler) design.

We also cite Colella et al. (2004) for comparing time series and spectrum.

2.       Datalogger: Line 90 to 92: I suggest to explain why only 3 channels are used.

Response: We designed the datalogger using three channels to record data in all spatial directions generally. For the particular case of 2D setups, where adjacent dipoles in the same direction are measured, it is also a good trade-off between minimizing the length of wires to the electrodes and the number of dataloggers to be installed in the survey area.

3.       Line 96: The GSM module is used only for the communication between the DL and the laptop as remote control…to download the acquired data the system uses a USB way. Why is it not possible to use a 3G module? I suggest to add one sentence to explain.

Response: There was no intention to download the big amount of data recorded over remote control. We just wanted to change parameters like gain and sample rate remotely and monitor small parts of the recorded time series to check the signal quality. Main aim was to minimize power consumption, which is significantly lower for the used GSM module compared to 3G- or 4G-modules.

4.      Line 120: I suggest to add on the figure 4 the frequencies indicated: powerlines, railway, the signals 0.2hz and the harmonics.

Response: We added the description of lines in the frequency spectrum to the modified Figure 4:

[Figure]

**Modified Figure 4: Exemplary voltage time series (a) from a square-wave current injection and frequency spectrum (b) as visible in the GUI monitoring software**

5.      Field case: Line 275-279: the authors wrote "Higher DD correspond to larger penetration depths but exhibit lower..."…the figure 14b show low S/N signals (blue color) in two zone (one shallow and one deep) with in the middle a better S/N signals zone. Therefore, the sentence needs some more details… it depends also for the electrical resistivity distribution. Low resistivity zone (i.e. clay) produce low S/N signals then relative high resistive layer (i.e. sandstone). I suggest to explain better this part.

Response: We agree that particularly in this example resembling huge resistivity contrasts there is a large dependency on the measured voltage and extended the description accordingly.

Changed text: "Figure 14b shows the obtained S/N for forward-directed standard dipole-dipole data in a so-called pseudo-section, i.e. as a function of the array midpoint and the dipole separation factor DD (dipole distance by dipole length) that indicates penetration depth. Generally, the S/N decreases with the dipole separation from about 0..20 dB for the shallowest penetrating data down to -10..-30 dB for the deepest data. A comparison with the very heterogeneous apparent resistivity (Figure 15) shows that S/N is strongly correlated with the measured voltages which are low above conducting zones."

6.      References: I suggest to check the matching between the list of the references and the indication in the text.

Response: Thank you for pointing this out. We checked in both directions and did the following corrections: i) The paper of Roßberg (2007) describing the low-frequency data logger used before was in the list but not cited in the text, which we now do in the introduction and when describing the logger design. ii) The citation Schünemann et al. (2007) is now given in the reference list. iii) The years of Johnson et al. (2001) and de Marcellis et al. (2012) were wrong and are corrected now.

7.      Figures: Figure 15: I suggest to add the unit (may be "m") for the X axis

Response: We have added units for x and y axis as suggested and rearranged the colorbar to be non-redundant:

[Figure]

Modified Figure 15: Calculated apparent resistivity (top) and normal reciprocity (bottom) for the three analysis approaches Lock-In (left), Stacking (center) and FFT (right).

Accordingly, Figure 14 was also changed:

[Figure]

Modified Figure 14: Signal-to-noise (S/N) values of field data: (a) Histogram compared to MSE of synthetic data, (b) pseudosection representation.

New references:

Balasco, M., Galli, P., Giocoli, A., Gueguen, E., Lapenna, V., Perrone, A., Piscitelli, S., Rizzo, E., Romano, G., Siniscalchi, A., Votta, M.: Deep geophysical electromagnetic section across the middle Aterno Valley (central Italy): preliminary results after the April 6, 2009 L'Aquila earthquake, Bollettino di Geofisica Teorica ed Applicata 52(3), 443-455, doi:10.4430/bgta0028, 2011.

Colella A., Lapenna V., Rizzo E.: High-resolution imaging of the High Agri Valley basin (Southern Italy) with Electrical Resistivity Tomography. Tectonophysics, 386, 29-40, 2004.

Rizzo E., Colella, A., Lapenna, V. and Piscitelli, S. High-resolution images of the fault controlled High Agri Valley basin (Southern Italy) with deep and shallow Electrical Resistivity Tomographies. Physics and Chemistry of the Earth, 29, 321-327, 2004.

Santilano A, Godio A, Manzella A, Menghini A, Rizzo E, Romano G. Electromagnetic and DC methods for geothermal exploration in Italy, state-of-the-art, case studies and future developments. First Break 33 (8), 81-86, 2015.

Schünemann, J., Günther, T., Junge, A.: 3-dimensional subsurface investigation by means of large-scale tensor-type dc resistivity measurements. Ext. abstract, 4th International Symposium on Three-Dimensional Electromagnetics, Freiberg, 2007.

Tamburriello G., M. Balasco, E. Rizzo, P. Harabaglia, V. Lapenna, A. Siniscalchi. Deep electrical resistivity tomography and geothermal analysis of Bradano foredeep deposits in Venosa area (Southern Italy): first results. Annals of Geophysics, 51 (1), 203-212, 2008.

---

## Referee Comment (RC2) · Anonymous Referee #2 · 5 Dec 2017

The introduction of the paper is well referenced and topics addressed clearly described.

As a comment : Complementary references may be find in the research domain civil engineering in the field of application of large scale structures monitoring.

Paragraphs dealing with the datalogger are clear.

Anyway, in its actual form it is difficult to evaluate what is the contribution part of authors on such system versus functionalities already proposed with these dataloggers by "Controlord" company.

Furthermore, it seems that the provider of such system has stop its activity at the end of 2016, so it should be of interest for the community to suggest alternative solutions.

Finally, synchronization of measurements seems addressed just by periodic adjustment of datalogger clock thank to GPS PPS. What is the time synchronization drift observed between datalogger and which accuracy is required for ERT measurements.

About the 3 post-processing approaches comparative study (FFT, Stacking and Lock-in), methodology and results obtained are clearly described and discussed.

In figure 9 RMS signal evolution analysis could be more commented versus DC and VPP for few parts that present some particular gap with global evolution of VPP and DC.

Did authors have addressed effect of time lag between synthetic data in their analysis? If yes is it integrated in the noise model used or will it be addressed in future works ?

For the field experiments, how many measurements repetition were made?

The conclusion is clear and perspective about CSEM should be moderate by the difficulty of time synchronization between dataloggers when high frequency analysis is required.

Remark : Page 9, line 289and 290 authors use "grey" comment in the text but figure 15 is in color.

---

## Author Comment (AC2) · 8 Dec 2017

Response to Anonymous Referee (gi-2017-37-RC2)

Dear Referee, Thank you for reviewing our manuscript and raising issues that help to improve the manuscript.

1. The introduction of the paper is well referenced and topics addressed clearly described. As a comment : Complementary references may be find in the research domain civil engineering in the field of application of large scale structures monitoring.

Paragraphs dealing with the datalogger are clear.

Response: Thank you very much. We did not look into the field of civil engineering as it is outside of the Geosciences scope that the journal is focused on. No change.

2. Anyway, in its actual form it is difficult to evaluate what is the contribution part of authors on such system versus functionalities already proposed with these dataloggers by "Controlord" company.

Response: Maybe we did not point this out clearly. As a major point, we added adjustable preamplifiers that allow changing the gain remotely during the experiment. Whereas the signals of most standard data loggers (e.g. temperature, water level, pressure etc.) remain largely constant in magnitude, the ERT signals cover a wide range of magnitudes as a function of different geometries (source-receiver distance) and thus voltage. Therefore, it is vital for good ERT data to optimize the input gain. Furthermore, the PPS synchronization was a feature not included in the Gigalog S. We added in the logger layout to make this clear: It is vital for good ERT data to optimize the input gain, as ERT signals cover a wide range of magnitudes due to different geometries (source-receiver distance). and The GPS timing can be transferred by an NMEA-Format string, which results in synchronization of one second accuracy, or by the PPS (Pulse-per-second) signal that is doing a synchronization within 1 ms.

3. Furthermore, it seems that the provider of such system has stop its activity at the end of 2016, so it should be of interest for the community to suggest alternative solutions.

Response: The company is still alive and selling the logger. In October 2017 we got a new firmware update (Version 1710) from Controlord. At any rate, one can take any basic logger to create such a data logger if it meets the required specifications. Giving potential names of manufactures would be arbitrary and therefore we would not like to give advantage to any of them.

4. Finally, synchronization of measurements seems addressed just by periodic adjustment of datalogger clock thank to GPS PPS. What is the time synchronization drift observed between datalogger and which accuracy is required for ERT measurements?

Response: The synchronization accuracy of the PPS signal is at around 1ms. The drift of the internal clock is below 20 ppm, typically 5 ppm. We observed a time difference between the dataloggers in the lower ms range. For ERT with periods above 1 s, the absolute timing is not important as the working point is determined by the processing software at any rate. Moreover, the mask technique makes the determination robust against time stretching.

5. About the 3 post-processing approaches comparative study (FFT, Stacking and Lock-in), methodology and results obtained are clearly described and discussed.

Response: Thank you.

6. In figure 9 RMS signal evolution analysis could be more commented versus DC and VPP for few parts that present some particular gap with global evolution of VPP and DC.

Response: You are right. We now discuss Figure 9 and the course of the individual curves in more detail. New text: The main criteria to obtain the real phase is the maximum of the DC function within a specific DC search area (THdc threshold of DC function, usually 25%). At an ideal square wave signal the DC-value of the convoluted signal is the desired amplitude. As this can lead to wrong results for signal contributions like overshoots, we additionally look if there is also a minimum of the Vpp function within a Vpp search area (THvpp threshold of Vpp function, usually 20%). The LabView function for the Vpp amplitude is calculating the positive and negative peak values from a histogram statistics that can cause the minimum of the Vpp function to be wide. To find the real phase within a wide Vpp minimum the minimum of the RMS function below the RMS search area (THrms threshold of RMS function, usually 10%) is used because the RMS value is very sensitive. This procedure is shown in the flow diagram in Figure 10 (a) and shown on the example in Figure 10 (b): The minimum of the quadratic sum

eval(i) of the 3 normalized criteria functions determines the phase, whereas we find the amplitude in the DC function DC(i).

7. Did authors have addressed effect of time lag between synthetic data in their analysis? If yes is it integrated in the noise model used or will it be addressed in future works?

Response: As explained above (4.), for the long-period ERT signals time lags are not an issue. The synthetic data have a length of 10min, which corresponds to the length of real ERT signals. Particularly the mask signal neglects the signal changes and is therefore robust.

8. For the field experiments, how many measurements repetition were made?

Response: We just did two "repetitions" by injecting two different current strengths that can be used to check repeatability. Furthermore, the use of reciprocal (forward and reverse) arrays allows for a more rigorous and well-established data check. One could of course also split the signal that contains about 100 periods into several segments and analyze them individually, thus deriving an additional standard deviation. We added some text: As another quality check we injected two different current strengths. The proportion of the recorded dipole voltages should be identical to the proportion of the injected currents. The small current does not saturate dataloggers in the vicinity of the current source, and high current provides enough amplitude for the biggest source-receiver distances.

9. The conclusion is clear and perspective about CSEM should be moderate by the difficulty of time synchronization between dataloggers when high frequency analysis is required.

Response: We are confident that the data loggers can be used for CSEM experiments. Actually we just finished a successful CSEM experiment with different sources and receivers. Although data analysis is still in progress, the absolute timing can be found by correlation methods. Also a systematic time dilatation can be compensated by appropriate time series analysis methods.

10. Remark: Page 9, line 289and 290 authors use "grey" comment in the text but figure 15 is in color.

Response: We specifically refer to the lower row of subfigures, where we used light grey and dark grey to indicate values above and below the color bar limit, respectively. Note that we provided a new version of Figure 15 in the response to reviewer 1 (AC1), which is as well in the new manuscript.